# Attitudes and adherence to changes in nutrition and physical activity following surgery for prostate cancer: a qualitative study

Luke A Robles ![ORCID] ,[1] Ellie Shingler,[1] Lucy McGeagh,[1,2] Edward Rowe,[3] Anthony Koupparis,[3] Amit Bahl,[4] Constance Shiridzinomwa,[5] Raj Persad,[3] Richard M Martin,[1,6] J Athene Lane[1,6]

RMM and JAL contributed equally.

**Correspondence to**
Professor J Athene Lane;
athene.lane@bristol.ac.uk

## ABSTRACT

**Objectives** Interventions designed to improve men's diet and physical activity (PA) have been recommended as methods of cancer prevention. However, little is known about specific factors that support men's adherence to these health behaviour changes, which could inform theory-led diet and PA interventions. We aimed to explore these factors in men following prostatectomy for prostate cancer (PCa).

**Design, setting and participants** A qualitative study using semistructured interviews with men, who made changes to their diet and/or PA as part of a factorial randomised controlled trial conducted at a single hospital in South West England. Participants were 17 men aged 66 years, diagnosed with localised PCa and underwent prostatectomy. Interview transcripts underwent thematic analysis.

**Results** Men were ambivalent about the relationship of nutrition and PA with PCa risk. They believed their diet and level of PA were reasonable before being randomised to their interventions. Men identified several barriers and facilitators to performing these new behaviours. Barriers included tolerance to dietary changes, PA limitations and external obstacles. Facilitators included partner involvement in diet, habit formation and brisk walking as an individual activity. Men discussed positive effects associated with brisk walking, such as feeling healthier, but not with nutrition interventions.

**Conclusions** The facilitators to behaviour change suggest that adherence to trial interventions can be supported using well-established behaviour change models. Future studies may benefit from theory-based interventions to support adherence to diet and PA behaviour changes in men diagnosed with PCa.

## BACKGROUND

Prostate cancer (PCa) is the most common form of cancer in men in the UK with over 48 000 new diagnoses every year.[1] Established risk factors are increasing age, ethnicity (black African or Caribbean) and a family history of PCa.[2] Modifiable factors, such as nutrition and physical activity (PA), have also

---

**STRENGTHS AND LIMITATIONS OF THIS STUDY**

⇒ This study provided a thematic analysis of men making diet and physical activity changes soon after prostatectomy, which included a negative case analysis to support the rigour of the study.

⇒ The study included a small sample size of 17 men.

⇒ Data analysis was limited by the lack of depth in men's responses, which is likely due to interviews being part of the data collection for a feasibility randomised controlled trial.

⇒ All men, except one, were white indicating that the study sample was not representative of the patient population.

---

been linked to PCa risk and progression.[3 4] A higher intake of cruciferous vegetables (eg, cabbage, cauliflower) is associated with a reduction in PCa incidence and progression.[5 6] Lycopene, a carotenoid found in many brightly coloured fruits and vegetables, has been linked with reduced risk of cancer progression post-diagnosis.[7] A high intake of dairy products is also associated with increased PCa risk.[8] With regard to PA, observational studies suggest that moderate to vigorous PA is associated with reduced risk of PCa-specific mortality and biochemical recurrence. More specifically, 3 hours of moderate to vigorous PA per week is associated with a 61% decrease in PCa mortality compared with less than 1 hour.[9] The increase of PA on lower risk of PCa-specific mortality and recurrence is supported by intervention studies.[10] In addition, PA has been shown to reduce adverse effects of treatment and improve quality of life, particular in men receiving androgen derivation therapy.[11 12]

The World Cancer Research Fund recommends making changes to nutrition and PA behaviours as methods of cancer prevention.[13] Such behaviour changes include maintaining

a plant-based diet (PBD) (ie, consuming more grains, beans,[14] five fruits and vegetables a day,[15] and performing 30 min of moderate to vigorous PA a day and limiting sedentary behaviours,[16] and the use of supplements, such as lycopene[17]). However, evidence has shown that most cancer survivors do not meet these recommendations. For example, Blanchard and colleagues[18] reported that, out of over 2000 PCa survivors, only 43% were meeting the recommendations for fruit and vegetable consumption and only 16% were meeting the recommendations for PA.

A systematic review[19] reported that nutrition interventions for cancer populations are rarely guided by behaviour theory. However, theory-based interventions were most effective at improving nutrition changes over a median follow-up of 12 months. There is limited evidence on psychological and behavioural factors that support adherence to nutrition interventions for men with PCa.[20] Furthermore, previous PA intervention studies with patients with chronic conditions, including cancer, have identified several factors that could support adherence to PA.[21] However, few of these studies have explored the psychological and behavioural factors which could align with existing models of behaviour change to enhance PA interventions in men undergoing prostatectomy. For example, a narrative review of behaviour change theories used in PA interventions in urological cancer survivors reported constructs of the Theory of Planned Behaviour and the Trans-theoretical Model have been shown to increase men's motivation to be more physically active either during or following PCa treatment.[22]

Our qualitative study aimed to identify factors associated with adherence to diet and PA interventions in men following prostatectomy for localised (organ-confined) PCa, which could inform such theory-led interventions in this patient population.

## METHOD

This descriptive qualitative study was part of a factorial randomised controlled trial (RCT), Prostate cancer Evidence of Exercise and Nutrition Trial (PrEvENT),[23 24] conducted at a single hospital in South West England. This trial assessed the feasibility and acceptability of nutritional and PA interventions for men after prostatectomy for localised PCa. Details of the trial can be found elsewhere (ISRCTN99048944).[23] In brief, men were randomly allocated to nutritional and/or PA interventions (table 1). This study was written in accordance with the Standards for Reporting Qualitative Research recommendations.[25]

### Participants

Seventeen men from the RCT, with an age range of 53–81 years (median=66 years), were recruited into the qualitative element of the study having provided informed consent to be contacted regarding an interview. Purposive sampling was employed to ensure maximum variation across the intervention arms and to ensure that the

**Table 1** Nutritional and physical activity interventions

| Intervention | Allocation* | Description |
|---|---|---|
| Nutritional | Plant-based diet | ▸ 5 fruits and vegetables per day <br> ▸ Substitute dairy milk for non-dairy alternative (eg, soya, almond or rice milk) |
| | Lycopene supplementation | ▸ 10 mg lycopene capsule taken once per day |
| | Control | ▸ No changes to usual nutrition |
| Physical activity | Brisk walking | ▸ 30 min brisk walking, 5 times per week |
| | Control | ▸ No changes to usual daily physical activity |

*Each participant was allocated to both a nutritional and physical activity intervention (factorial randomisation).

sample consisted of various demographic characteristics such as age, employment status and educational level.[14] Trial eligibility included men who were diagnosed with localised PCa, undergoing prostatectomy with no restrictions to performing the interventions. Twenty-five men were approached for interview in person during their 6-month research clinic visit appointment of the feasibility study. Six men were unable to attend due to external and personal circumstances (ie, did not have the time during the clinic appointment (n=3), interviewer not available (n=2) and participant unwell (n=1)) and two men declined giving no reason. Seventeen men agreed and were interviewed. All men interviewed, except one man who was Caribbean, were reported as white British or white other. Most men were retired (n=12) and married (n=13). Over half of the men were educated to secondary school level (n=9) (table 2).

### Data collection

Men took part in semistructured interviews between April 2015 and May 2016 after completing their final 6-month follow-up. Interviews were conducted in person within a private research clinic room (n=12). For those who were unable to attend in person, a telephone interview was arranged (n=5). Interviews were conducted by three authors (ES, n=9; LM, n=7; LAR, n=1), whose backgrounds include public health (ES) and health psychology (LM, LAR), and lasted between 19 and 84 min. All three authors were involved in the data collection process of PrEvENT, although they had very minimal contact with participants. Interviews followed a predefined interview topic guide (online supplemental material 1), in which questions focused on participants' experiences of performing the interventions from a trial perspective. However, participant responses often included topics associated with

**Table 2** Participant characteristics and intervention allocation

| | | n=17 | |
|---|---|---|---|
| | | n or median | % (range) |
| Age (years) | | 66 | 53–81 |
| Ethnicity | White British/white other | 16 | 94 |
| | Caribbean | 1 | 6 |
| Marital status | Married | 13 | 76 |
| | Not married | 4 | 24 |
| Education level | Secondary school (eg, O-levels, GCSE) | 9 | 53 |
| | University | 7 | 41 |
| | Further education (eg, A-levels, HND) | 1 | 6 |
| Occupation status | Retired | 12 | 71 |
| | Employed | 5 | 29 |
| Intervention arm | Lycopene and brisk walking | 4 | 23 |
| | Lycopene and physical activity control | 3 | 18 |
| | Plant-based diet and brisk walking | 3 | 18 |
| | Plant-based diet and physical activity control | 3 | 18 |
| | Brisk walking and nutritional control | 3 | 18 |
| | Control | 1 | 6 |

O-level, GCSE: national school examinations at age 16 years. A-level: national school examinations at age 18 years. Higher national diploma (HND) is a work-related course provided by higher and further education colleges in the UK. A full-time HND takes 2 years to complete and generally is the equivalent to 2 years at university.

long-term adherence to behaviour changes. One man in the control group was included in the sampling. He received no intervention aside from standard publicly available nutrition and PA information, if requested. Data from responses about his diet and PA before participation in the trial were only used for analysis.

## Data analysis

Interviews were audio recorded and transcribed for analysis. The transcripts were checked against the audio recordings for accuracy. Data were analysed using inductive thematic analysis with the aid of NVivo V.10 software.[26] This method of analysis was chosen with the aim of understanding participant experiences of making behaviour changes beyond those related to study processes of the RCT (eg, feasibility outcomes). There were also no preconceptions about what themes would be identified from the data. Data analysis involved reading through the transcripts to increase familiarity with the data. They were, then, coded for items of data relating to the research question. The coding process was performed by one researcher (ES) and checked for consistency by a second researcher (LAR). These codes were collated to form themes, which were reviewed and refined until a coherent narrative of the men's experiences was produced. Themes were reviewed and discussed regularly by both researchers to ensure they accurately represented the original data. A constant comparative approach was used to look at differences between sample characteristics, such as age, employment status and intervention arm. Negative case

analysis (ie, identifying contradictory data) was used to broaden or confirm the interpretation of the themes and was resolved through discussion between the researchers and revisiting the transcripts. Data analysis was conducted throughout the data collection process to allow for initial themes to be explored in subsequent interviews. This also allowed researchers to decide when data saturation (ie, no new themes or additional information emerged from the interviews) was reached.

## Patient and public involvement

A PCa patient and public involvement group were involved in the concept stages of PrEvENT and reviewed trial documentation, including the interview participant information sheet, consent form and topic guide.

## RESULTS

The analysis yielded five overarching themes: (1) causal beliefs about PCa; (2) perceptions of a healthy diet and PA before diagnosis; (3) barriers to adherence; (4) facilitators of adherence and (5) perceived benefits of behaviour change. The thematic map is shown in figure 1.

## Causal beliefs about PCa

Men perceived that cancer was caused by external factors such as ageing, genetics and environment agents (ie, radiation from nuclear sites). When asked about the impact of diet and PA with cancer, several men believed there was little or no association. Men obtained information about

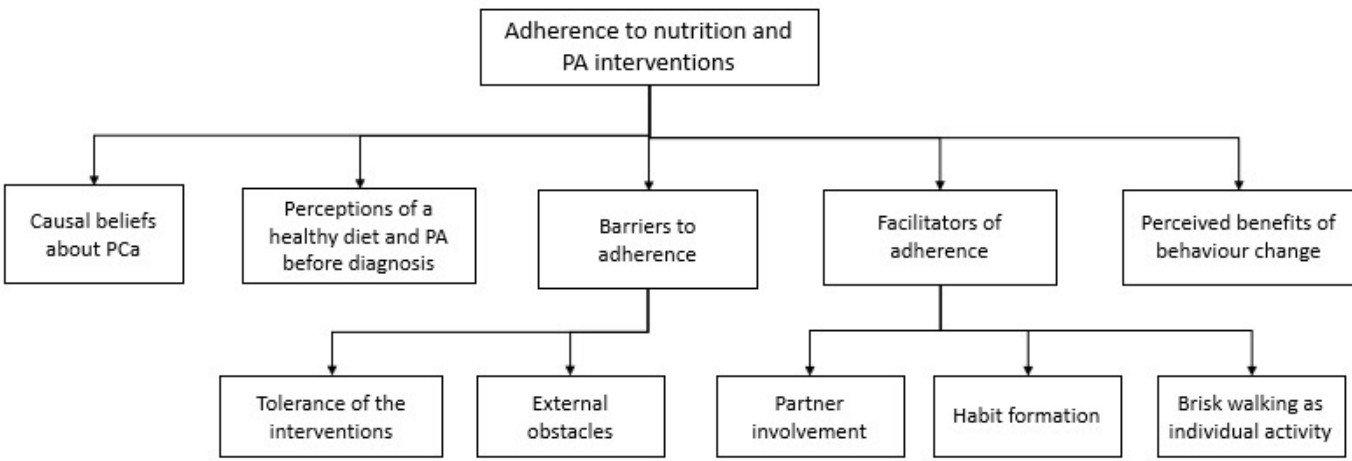

**Figure 1** Thematic map of qualitative analysis. PA, physical activity; PCa, prostate cancer.

PCa from media sources that were, at times, found to be conflicting.

> …I've looked at these things [causes of cancer] to some extent and I must admit that the evidence for diet-cancer links, to my view, has been weak. (P6, PBD and brisk walking)

> Well you read it in the paper and sometimes you think there might be [a link with cancer]. (P8, lycopene and brisk walking)

In contrast, a small number of men reported that they believed that healthy eating and regular exercise were associated with their cancer and this was one reason for maintaining a healthy diet and being physically active.

> Interviewer: …before you took part in the trial, had you ever thought about the links between your lifestyle, what you ate and how much activity you did and cancer?

> Participant: Well I was concerned that it might be related so I have always tried to eat the right things and do exercise and walking so I just carried on as before. I didn't do any extra walking but I do try and walk at least two miles a day. (P13, PBD)

### Perceptions of a healthy diet and PA before diagnosis
Men across all the intervention arms believed that they maintained a healthy diet before being diagnosed with PCa. However, the evidence for this notion was mixed. Some men described being able to effectively maintain a healthy diet.

> I really, sort of, eat a fairly Mediterranean diet. I use olive oil instead of butter, for example. If I have a sandwich or something, I put olive oil on. We cook all our own vegetables. I used to have an allotment, which I had to give up because of my leg, because of my knee. (P5, lycopene and brisk walking)

However, other men described making extensive changes on starting the PBD within the trial.

> …I found that I was really, sort of, [toning] myself up almost on fruit and veg. I think it said you had to eat 5 more portions of fruit and veg a day than normal, so I was getting up to, at some stages, about 20, I think, a day. (P17, PBD and brisk walking)

Men generally described themselves as participating in daily PA, such as going out for regular walks, before being diagnosed. Some men also belonged to a gym.

> I tend to do stretching exercises every day and I do a lot of gardening as well. I love gardening and I do walk. As I say, I've got two little terriers. (P15, lycopene)

> I'm quite active anyway. Even beforehand I'd get a bike ride, a good two hour or so bike ride once a week and a gym session and once or twice round that walk anyway or sometimes longer. It wasn't a complete change of lifestyle for me. (P14, lycopene and brisk walking)

### Barriers to adherence
#### Tolerance of the interventions
Few obstacles were identified by men regarding their ability to adhere to the interventions. Men with comorbidities, including knee pain, were restricted from walking 'briskly' as this was found to aggravate their physical conditions. Some men struggled to adhere to soya milk, mostly due to its taste in coffee or tea. One man believed lycopene caused him some constipation and, therefore, he preferred not to consume his supplements in the long term.

> Interviewer: If we said, 'Can you do this for 12 months?' Could you have carried on?

> Participant: I would have done yes, but as choice I would say no. I do believe that it causes me slight constipation so I would rather not. (P10, lycopene)

#### External obstacles
Most men relied on good weather. There was also little motivation to walk elsewhere when the weather was bad.

…there were some day where it was a total wash-out, and you think, 'Well, there's no point in even trying,' you know. 'I'll make this my quiet day'… (P3, brisk walking and lycopene)

There were also clear differences in men's perceived ability to adhere to brisk walking between those employed and those retired. Work was described as affecting men's success at maintaining their brisk walking.

Anyway, before I went back to work, it was easy to discipline myself to say, 'Right, I'm going to go walking in the morning and in the afternoon, twice a day,' but when I went back to work, that wasn't so easy. (P6, PBD and brisk walking)

I did think if I'd been working, especially over the winter, it would have been quite difficult to do because you get up and go into work in the dark and come home in the dark. (P14, lycopene and brisk walking)

Activities that intervened with men's usual routine, such as going on holiday, eating out and staying with friends, were reported to affect some of the men's adherence to both the PBD and brisk walking interventions.

… I went to my son's [place] and they don't eat a lot of fruit and vegetables there right now, so perhaps for a couple of days then, it was a low count. (P2, PBD)

… we were travelling, visiting friends and doing things, so there were some days there when I just couldn't do any walking. (P1, brisk walking)

### Facilitators of adherence
#### Partner involvement
Men often suggested that their wives or partners would frequently prepare their meals and this would help them with their adherence to the PBD, especially if they also consumed a diet high in fruits and vegetables.

She is wonderful and she looks after me absolutely, 100%, our food is ready by six…My wife is a three veg, four veg, five veg and, she is greens, she thinks they are wonderful. (P2, PBD)

#### Habit formation
Five out of the six men in the lycopene arm were on prescribed medication for other health conditions. They suggested that the routine of self-medicating meant that they found it easy to adhere to taking the supplements.

It becomes very easy, because the Lycopene, I took every morning with my hypertension medication and it just became part of the breakfast … (P3, lycopene and brisk walking)

A couple of brisk walking men, who were physically active and belonged to a gym prior to initiating behaviour changes for the purpose of the trial, mentioned that they would overcome barriers, such as bad weather, by incorporating it as part of their usual indoor exercise routine.

…I built my walk into the gym routine. I did 30 minutes on a treadmill sitting at about 6 kph or something like that with grading… (P14, lycopene and brisk walking)

#### Brisk walking as individual activity
Although attempts were made by men to carry out their brisk walk with others, most men claimed that they were happy to walk by themselves and were not dependent on others to help motivate them. Men discussed that one of the reasons why they brisk walked by themselves was due to its intensity (ie, walking at a pace where they could talk but not sing) as they felt others were not able to walk at the same pace.

My wife has joined a walking group, but they don't go fast enough. It was too much of an amble. She doesn't walk very fast, by comparison. If ever we're going anywhere, I have to modify my pace to suit her. It was better to do it on my own. (P1, brisk walking)

### Perceived benefits of behaviour change
Most men reported there were many benefits to being more physically active. Several men discussed that going out for a walk provided them with a structured way of performing a reasonable level of PA, which they would not normally do.

I think, I mean, although I've painted a picture of being quite active, then you know, I mean, it wasn't very organised, you know what I mean? What this did was to impose a routine on me, which I was quite happy with. And it's like setting a goal, isn't it? (P3, lycopene and brisk walking)

It also gave them a sense of feeling healthier. One man spoke about how walking to work enabled him to 'clear his head' before starting work. Another man even associated his brisk walking to success of subsequent radiotherapy.

The walking because it kept me a bit healthier and fitter I think I did better on the radiotherapy. (P8, lycopene and brisk walking)

Men did not comment on the physical or psychological outcomes they experienced from consuming more fruits and vegetables or lycopene despite knowing the potential health benefits.

### DISCUSSION
#### Summary of findings
We aimed to explore factors influencing adherence to nutrition and PA interventions in men, who had prostatectomy following a diagnosis of localised PCa. Our findings showed that men believed their cancer was caused by external factors, such as age and genetics. They discussed eating healthily and regularly exercising before their diagnosis and barriers and facilitators to their behaviour

changes. Overall, men found the PA intervention was beneficial to their health and well-being.

## Support with other studies

Men were not fully convinced that cancer was caused or related to their nutrition or PA. They attributed the cause of their cancer to external factors including age and genetic factors. These findings are supported by another qualitative study,[27] which showed that PCa survivors can overestimate the significance of environmental factors, such as pollution and stress, and underestimate behaviour factors associated with increased cancer risk, such as obesity and inactivity. In contrast, observational evidence showed that a high proportion of women attributed diet (68%) to their breast cancer diagnosis in addition to external factors (ie, hormones).[28] Furthermore, in a sample of 40 men interviewed about their lifestyle behaviours following their PCa diagnosis, 60% were obese and 88% were not motivated to change their smoking, alcohol and/or their eating behaviour.[29] These findings could be indicative of men's preference to believe in causal factors that are outside their control, and reinforce the importance of lifestyle interventions at the time of diagnosis.

Men from all the intervention groups believed that they adhered well to their nutrition intervention. While some men followed the intervention guidelines, others made quite extreme changes to their diet, such as eating well over the recommended daily intake of fruits and vegetables. This suggests that men may benefit from more education on eating practices, including more detail on portion sizes.

Men's tolerance to changes in their diet impacted on their adherence to their nutrition interventions. Some men did not like the taste of soya milk and reverted to dairy milk or alternative forms of dairy-free milk. This somewhat contradicts findings from previous trials that have shown men to adhere well to a daily consumption of soya products over significant follow-ups. However, these trials incorporated soya products in the form of drink supplements[30 31] and soya bread.[32] Thus, the way in which soya products are consumed could influence how men adhere to these products in the long term. With regard to lycopene, its side-effects are not well known although other PCa trials have reported diarrhoea and flatulence as plausible side-effects of the supplement in few cases.[33] One man did believe that the constipation he experienced during the trial was due to taking lycopene. Therefore, constipation could potentially be a side-effect and men would need to be aware of these effects and advised on how to manage them in future trials.

Barriers to brisk walking included weather conditions and a lack of time. These barriers to regular walking have been cited by prostate and other patient populations.[29 34 35] Men were assessed for comorbidities prohibiting them from performing brisk walking before entering PrEvENT. Therefore, it could be speculated that the physical restrictions to brisk walking reported by men are indicative of their motivation to brisk walk when obstacles arise.

Partners were found to be significantly involved in choosing and preparing meals for men. Partners are often involved at each stage of men's treatment pathway, including helping them comply with pre-prostatectomy preparation, such as improving fitness and losing weight.[36] Thus, this finding suggests that men would adhere better to PBD interventions with partner involvement. In contrast, men discussed the PA intervention as one which they preferred to do by themselves. This finding somewhat contradicts evidence which has shown men to report physical, mental and relationship benefits from PA interventions involving their partners.[37] However, the prescribed nature of couple-based interventions is likely to attribute to its efficacy.

A facilitator to the lycopene intervention was men consuming the supplement along with their existing medication regimen. In a similar vein, men who were exercising regularly, before being enrolled in the RCT, were able to include brisk walking into their exercise schedule. Evidence from a previous RCT suggested that men's exercise adherence was more difficult for those who had not considered exercising before entering the trial.[38] These facilitators for lycopene and brisk walking adherence suggest adherence is linked to habitual behaviours (ie, actions to contextual cues). Habitual formation behaviours have been shown to increase adherence to both nutrition and PA interventions. The current findings suggest incorporating new health behaviours with existing healthy habits could strengthen adherence.[39]

Physical and psychological benefits were reported by those men who brisk walked. These beneficial effects have been reported in a prospective study measuring PA in men with PCa.[40] Men who adhered to 150 min of moderate PA post-diagnosis had significantly better physical ($\beta$=6.01, 95% CI: 4.15 to 7.86) and mental ($\beta$=2.32; 95% CI: 0.29 to 4.34) quality of life (ie, physical functioning and better mood states) compared with men who were non-adherent. Such physical and psychological outcomes have been reported as facilitators to exercise[29] and have the potential to help men adhere well to their brisk walking.

## Strengths and limitations

The strength of this study is that it has provided a thematic analysis of men making diet and PA changes soon after prostatectomy, which included a negative case analysis to support the rigour of the study. However, this study has several limitations. Data analysis was limited by the lack of depth in responses from the interviews. This is likely due to the interviews being part of data collection for a feasibility RCT, which assessed trial processes as well as intervention adherence. Therefore, further assessments of rigour would not have benefitted the data analysis. The sample size was small (n=17) and all men, except one, were white and the majority were married. It is unlikely that the data fully represent the

experiences of men from other ethnic groups or single men without support from partners with their intervention. In addition, men in all the intervention arms discussed that they were already maintaining a healthy diet and engaging in regular PAs before their diagnosis. This could suggest that the current findings are limited to men more willing and able to perform these health behaviours. As this is a qualitative study, findings are based on subjective accounts of behaviour change and there is the chance of men over-reporting areas of their behaviour change due to recall bias and men wanted to please the researchers.[41]

### Main implications and future research

Intervention studies should embrace the use of social support to reinforce adherence to dietary changes, especially with PBD interventions where partners are involved with meals prepared at home. Behavioural interventions that can be performed with existing behaviours (eg, medication regimen) are likely to increase participants' confidence and adherence. Further work may want to tailor interventions that consider contextual cues and one's belief in the ability to perform the desired behaviour, as well as behavioural strategies that support adherence. A theory-led behavioural model can both guide and assist with evaluating interventions.[42] Our study findings indicate that men are motivated to make changes to their diet and level of PA following prostatectomy. However, men's motivation was not related to beliefs that diet and PA were associated with their PCa. Other psychological factors could explain men's motivation to adherence to these behaviour changes, such as symptom control, which could be explored using qualitative studies. Barriers to adhering to their behaviour changes related to physical (ie, weather, time) and social opportunities (eg, going on holiday). These findings suggest that future nutrition and PA interventions guided by a behavioural model, which help identify these barriers and incorporate techniques such as problem-solving, will improve adherence.[43 44] The Capability, Opportunity, Motivation-Behaviour (COM-B) model[45] could be one that is suitable for this patient population. This model proposes that a person's motivation to perform and maintain a behaviour is supported by their capability (ie, psychologically and physically) and opportunity (ie, social and physical) to perform the behaviour. Future studies may consider exploring the use of this model in nutrition and PA intervention studies with PCa populations.

### CONCLUSION

The findings from this study may be helpful in developing and implementing future nutrition and PA interventions in men after receiving prostatectomy for PCa. This qualitative study suggests that behaviour change models could support adherence to nutritional and PA behaviours.

**Author affiliations**
¹NIHR Bristol Biomedical Research Centre, University Hospitals Bristol and Weston NHS Foundation Trust and University of Bristol, Bristol, UK
²Supportive Cancer Care Research Group, Faculty of Health and Life Sciences, Oxford Institute of Nursing, Midwifery and Allied Health Research, Oxford Brookes University, Oxford, UK
³Bristol Urology Institute, Department of Urology, North Bristol NHS Trust, Bristol, UK
⁴Bristol Haematology and Oncology Centre, University Hospitals Bristol and Weston NHS Foundation Trust, Bristol, UK
⁵Clinical Research Centre, North Bristol NHS Trust, Bristol, UK
⁶Bristol Medical School, Population Health Sciences, University of Bristol, Bristol, UK

**Acknowledgements** The authors would like to thank all participants for taking part in the qualitative interviews. The authors would also like to thank Dr Aidan Searle for his advice on the qualitative analysis of this study.

**Contributors** ES and LM collected the data. LAR, ES and LM analysed and interpreted the data and were major contributors to the manuscript. LM, AB, RP, JAL and RMM conceived the study, reviewed the article and critically revised the manuscript. ER, AK, AB, CS and RP were involved in the study conduct and critically revised the manuscript. All authors read and approved the final manuscript. JAL is the guarantor.

**Funding** This study was funded by the National Institute of Heath and Care Research Bristol Biomedical Research Centre.

**Disclaimer** The views expressed are those of the author(s) and not necessarily those of the NIHR or the Department of Health and Social Care. The funder was not involved in the design of the study and collection, analysis, and interpretation of data and in writing the manuscript.

**Competing interests** None declared.

**Patient and public involvement** Patients and/or the public were involved in the design, or conduct, or reporting, or dissemination plans of this research. Refer to the Methods section for further details.

**Patient consent for publication** Not required.

**Ethics approval** This study involves human participants and PrEvENT received Research Ethics Committee approval from the National Research Ethics Service Committee South West–Cornwall & Plymouth on 8 April 2014 (REC ref 14/SW/0056). All participants provided written informed consent on enrolment into the trial. Additional oral consent was obtained from participants who took part in telephone interviews. A participant information sheet and consent form were sent in advance of the telephone interviews.

**Provenance and peer review** Not commissioned; externally peer reviewed.

**Data availability statement** Data are available upon reasonable request. The datasets used and/or analysed during the current study are available from the corresponding author on reasonable request.

**ORCID iD**
Luke A Robles http://orcid.org/0000-0003-2882-9868

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
