## [Reviewer comments · BMJ Open]

ARTICLE DETAILS

TITLE (PROVISIONAL)	ATTITUDES AND ADHERENCE TO CHANGES IN NUTRITION AND PHYSICAL ACTIVITY FOLLOWING SURGERY FOR PROSTATE CANCER: A QUALITATIVE STUDY
AUTHORS	Robles, Luke; Shingler, Ellie; McGeagh, Lucy; Rowe, Edward; Koupparis, Anthony; Bahl, Amit; Shiridzinomwa, Constance; Persad, Raj; Martin, Richard; Lane, Athene

VERSION 1 – REVIEW

REVIEWER	Dobova, Svetlana Mexican Social Security Institute, Epidemiology and Health Services Research Unit, CMN Siglo XXI
REVIEW RETURNED	28-Jul-2021

GENERAL COMMENTS	The manuscript entitled “Attitudes and adherence to changes in nutrition and physical activity following surgery for prostate cancer: a qualitative study” aimed at investigating factors that increase men’s adherence to the nutrition and physical activity interventions. I find it difficult to understand the methods, results, conclusions, and scientific contribution of this study. First, the introduction section should specify if the study was informed by a theory or conceptual framework? If so, describe it and how it was used. If not, say so and why. It is confusing that the authors mentioned the Theory of Planned Behaviour only in the discussion section. Second, the study design is not clearly defined. The authors mentioned that they performed a qualitative study as part of a secondary analysis of a factorial randomised controlled trial. Yet, there are several types of qualitative studies (See articles about this design written by M. Sandelowski in 2000 and 2010 in the journal Research in Nursing & Health). Therefore, the study type should be clearly specified. Third, it is unclear, what is the authors' argument for thematic analysis in the tradition of Braun and Clarke? Why is this method suited to the study at hand? Forth, there are discrepancies between the study objectives and study methodology. The study objectives focus on the factors that supported and hindered men's adherence to the nutrition and physical activity; yet, in the method section on page 4 the authors specified that the participants “were asked mainly about their diet and PA before participation in the trial”. At the same time, the patient interview guide that the authors presented focuses mostly on the participants' experiences with the intervention, asking about the positives and negative elements of the intervention arm and participant’s opinions about the associations between diet, physical activity, and cancer. There is unclear how the questions in the interview guide as shared in the paper helped answer the questions
--

posed in the objective.

Fifth, at what level of abstraction did the authors aim in identifying themes? As I read all of them, each is purely descriptive and thus generally best classed as categories and not themes. Refer to the editorial, "Confusing Categories and Themes" by J. Morse (2008). Might you revise your level of analysis? I also urge you to reconsider whether you have sufficient data to report certain themes. Most themes are described with a thin narrative and few quotes. The balance of narrative to data is typically indicative of the depth of development and representation of richness in the underlying data. Do the authors think they've adequately captured what the participants and data convey? I'd like to see a good deal more development here.

Sixth, it is important to include in the method and the discussion section the information on how the rigor of the study was established. There are a number of references for this, including the seminal text by Lincoln & Guba (1985) and the article, "Critical Analysis of Strategies for Determining Rigor in Qualitative Inquiry" by J. Morse (2015).

Seventh, there are multiple discrepancies between the statements (or the names of the categories) and quotes presented to support the statements/categories in the results section. For instance, the author wrote on page 6: "While other men described making extensive changes on starting the PBD within the trial." However, the quote that follows this statement is "As I say, I used to eat an awful lot of fruit and vegetables beforehand..." Another example: the authors put the following statement as an example for the "physical limitations" category: "Most men also relied on good weather. There was also little motivation to walk elsewhere when the weather was bad. "...there were some days where it was a total wash-out, and you think, "Well, there's no point in even trying," you know. "I'll make this my quiet day"..." However, it seems that the above-mentioned example is more appropriate for the "external obstacles category." Furthermore, the results section is difficult to understand as it lacks a clear picture overall. Therefore, the result section will benefit if the thematic map of the study is presented.

Eighth, how are the authors constructing the discussion for this manuscript? Typically, a strong discussion recapitulates the main findings in brief and then goes on to argue original contributions, areas where findings corroborate extant literature, where the findings conflict, and finally limitations. This progression allows the authors to draw clearly defined conclusions in line with the method and results and to outline implications. I do not see this structure in your discussion. Critically, many statements in the discussion section are not supported by the study result. For instance, the following statements in the discussion section are not supported clearly by the study results: (1) "The findings suggest that a diagnosis of cancer can provide an opportunity for men to make changes to their nutrition and PA." (2) "While some men followed the intervention guidelines, others made quite extreme changes to their diet, such as eating well-over the recommended daily intake of fruit and vegetables." (3) "The facilitators to change identified in this study (i.e. ... self-efficacy with taking lycopene and exercising)"; (4) "This qualitative study suggests that behaviour change models could help both inform interventions and promote long-term adherence to nutritional and PA behaviours."

Ninth, the "strengths and limitations" section should be revised. First, it is unclear how the study's strength could be its "findings on the psychological, behavioural, and social factors associated with adherence to men's changes to their diet and PA". Usually, the

	strength of the study should be judged on the rigor of the study methodology; therefore, the researchers should specify on what quality criteria they based their research. Please see, Lincoln & Guba (1985) and the article on "Critical Analysis of Strategies for Determining Rigor in Qualitative Inquiry" by J. Morse (2015), as well as, Treharne GJ, Riggs DW. Ensuring quality in qualitative research. In: Rohleder P, Lyons AC (eds) Qualitative Research in Clinical and Health Psychology. Basingstoke: Palgrave Macmillan, 2014; pp. 57–73. Furthermore, although in the result section the authors mentioned that “men across all the intervention arms believed that they maintained a healthy diet before being diagnosed” and the authors provided several quotes that described the men’s healthy behaviors before the study, they did not mention this fact in the limitation section. However, it is well known that the new health-related behaviors’ is easier to achieve in people with previous healthy behaviors compared to those without previous health behaviors; therefore, the factors that supported and hindered men’s adherence to the nutrition and physical activity could be different among those with and without previous healthy behaviors.
--	---

REVIEWER	Crevena, Richard Medical University of Vienna
REVIEW RETURNED	29-Dec-2021

GENERAL COMMENTS	To my opinion, the manuscript (bmjopen-2021-055566, "ATTITUDES AND ADHERENCE TO CHANGES IN NUTRITION AND PHYSICAL ACTIVITY FOLLOWING SURGERY FOR PROSTATE CANCER: A QUALITATIVE STUDY ") could be of interest of the reader of your journal, the BMJopen. Importance of the question studied/described: adequate Originality of work: adequate Appropriateness of approach and experimental design: qualitative study using semi-structured interviews Clarity of writing and soundness of organisation of the paper: adequate Soundness of conclusion and interpretation and relevance of discussion: not adequate Reference list: adequate
---

REVIEWER	Costi, Stefania University of Modena and Reggio Emilia
REVIEW RETURNED	30-Dec-2021

GENERAL COMMENTS	Comments General comments The present manuscript is a qualitative study with the aim to explore factors that increase men’s behavioural intentions and support their adherence to a diet and PA intervention. I think it could be of interest by readers of BMJ Open. However, there are some points, that I would ask you to clarify. Special comments Abstract
---

Page 4, line 12: I suggest writing the word physical activity in full for the first time and then using the abbreviation “PA”.

Page 4, line 14: Could you explain what are the “theory-led interventions”?

Page 4, line 30-31: Are these barriers physical limitations (e.g. pain, constipation)? I suggest to specify what kind of nutrition and PA limitations and the external obstacles that men reported as a barriers.

Main text
Introduction

Page 5, line 7: I suggest to expand the rationale for PA and cancer survivorship. Exercise is stronger recommended as a strategy to prevent the side effects of PCa treatments and improve patients’ quality of life. Two recent systematic reviews discussed the role of exercise in patients with prostate cancer receiving androgen deprivation therapy about its effectiveness, feasibility and safety (Bressi et al. Physical exercise for bone health in men with prostate cancer receiving androgen deprivation therapy: a systematic review. *Support Care Cancer*. 2021, doi: 10.1007/s00520-020-05830-1. Cagliari M et al, Feasibility and Safety of Physical Exercise to Preserve Bone Health in Men with Prostate Cancer Receiving Androgen Deprivation Therapy: A Systematic Review).

Methods

Page 5, line 41: Was the study prospectively recorded in one of the available public databases?

Page 6, line 12: How were participants approached (e.g. face-to-face, telephone, email...)?

Page 6, line 12: Could you specify the “external circumstances” of the six men unable to attend?

Page 6, Table 2: Could you specify in the table legend what means “further education”, with examples?

Page 6, line 45: What was the duration of the interviews?

Page 6, line 46: Why did you decide to not include focus group to encourages respondents and interaction with each other, and to share perspectives?

Page 6, line 49: What were the researchers credentials? What experience or training did the researchers have?

Page 6, line 51: By whom and how was the questionnaire written (Supplementary material 1)? Were questions, prompts, guides provided by the authors?

Page 6, line 51: What methodological orientation was used (e.g.

	grounded theory, discourse analysis, ethnography, phenomenology, content analysis)? Page 6, line 53-54: I understand that these sentences referred to the man (n = 1) included in the control group of the RCT Prostate cancer Evidence of Exercise and Nutrition Trial (PrEvENT), but the subject is plural. Please clarify. Results Page 7, line 39: I suggest to specify only the first time that PBD is the Plant-based diet, and then use the acronym consistently. Page 7, line 23: It could be of interest a description of some theme of the man included in the control group of the PrEvENT study. Discussion Page 11, line 32: In the literature the role of partner is considered a usefull support for men with PCa to adopt and maintain healthy behaviours. Partner may influence each other's diet and also exercise behaviours (PMID: 25807856). Your findings did not confirm the role of partners in supporting PA, so I suggest to discuss this point.
--	---

REVIEWER	Edmunds, Kim Griffith University, Centre for Applied Health Economics
REVIEW RETURNED	08-Jan-2022

GENERAL COMMENTS	Thankyou for the oppportunity to review this paper. Lifestyle compliance and adherence are important considerations for cancer patients and it is pleasing to see qualitative investigation happening alongside randomised clinical trials. I thoroughly enjoyed the paper and encourage you to extend your research in this important area as you propose in your protocol. While much research has been carried out on adherence, PCa patients with early stage disease are a unique group because they tend to be relatively healthy and the evidence for the impact of lifestyle change on their disease is strong. Any research that interrogates the why behind adherence or otherwise is always welcome. That said, there are areas in your paper I believe could be strengthened. Most importantly is that much of the research that backgrounds your study tends to be quite dated. It gives the reader the impression that you are not up to date with the latest developments in exercise oncology. International leaders in this area in PCa are Robert Newton Daniel Galvao, Dennis Taaffe, Kerry Courneya. Other exercise oncologists of international note are Kathryn Schmitz, Anna Campbell, Sandi Hayes, and others. Thank you for incorporating a PPI group in your research. I comment on the paper as and where such issues arise. Abstract Page 4 Line 22-expression and accuracy Participants were 17 men with a median age of 66 years,,who underwent surgery
---

	Why don't you use prostatectomy rather than surgery, as you did in your feasibility study? I would think it more appropriate for your readership at BMJ. Line 28-expression relationship of nutrition and PA with PCa...OR the impact of nutrition and PA on PCa... Page 5 Line 5-expression There is also evidence that high intake of... Line 9ff updated references Many of the references cited are somewhat dated and while Kenfield is a seminal article, systematic reviews by internationally recognized exercise physiologists (e.g. Cormie et al. 2017 The impact of exercise on cancer mortality...) have updated these and are worth including. Line 19 expression ...were meeting the recommendations for fruit... OR rewrite ...out of 2000 PCa survivors, only 43% were meeting recommendations for fruit and vegetable consumption and only 16% (were meeting recommendations) for PA. Line 27 This reference is 2011. Much has been written since about this area re motivation, compliance and adherence in exercise oncology. Courneya, K. did much of the early work. A recent umbrella review which includes seven exercise oncology papers is informative in terms of all identifying the key factors (Collado-Mateo 2021 Key factors associated with adherence to physical exercise..). I believe you need to situate your study within this more recent literature. Page 6 Line 15 redundant language ...reported as White British... Line 49 consistent use of numbering ...All three authors... Page 10 Line 6 expression ...others, most men claimed/admitted that.., More recent research needs to be incorporated in your discussion. This is where you compare what you found with what others have found, so ideally, should be the most recent research in the area. Page 11 Line 28 verb form consistency ...three constructs: having a positive attitude to a behaviour, (2) perceiving others to be supportive of it, and (3) believing... Line 57 ...another study
--	--

	Page 12 Line 3 ...another qualitative study... Interestingly, both the findings from these studies seem to suggest that these men prefer to blame things outside their control or not take responsibility for their actions (environment and work/life stress, vs overeating and being inactive?? Perhaps this could be used in devising motivational change...??? Just a thought I am not sure I agree with your tentative finding that men would adhere to healthy changes regardless of their perceived risk. Perhaps something else is happening that fits more with their preference not to take responsibility...if some one organises it and tells them what to do, they will happily do it??? Just a thought I think the small sample size is a limitation that should be mentioned. Congratulations on a well conducted study on an important topic. My suggested changes are minor. However, I believe it is important in the background and in the discussion to incorporate other more recent studies against which to compare your own.
--	--

VERSION 1 – AUTHOR RESPONSE

Reviewer: 1

Dr. Svetlana Doubova, Mexican Social Security Institute

Comments to the Author:

The manuscript entitled “Attitudes and adherence to changes in nutrition and physical activity following surgery for prostate cancer: a qualitative study” aimed at investigating factors that increase men’s adherence to the nutrition and physical activity interventions. I find it difficult to understand the methods, results, conclusions, and scientific contribution of this study.

First, the introduction section should specify if the study was informed by a theory or conceptual framework? If so, describe it and how it was used. If not, say so and why. It is confusing that the authors mentioned the Theory of Planned Behaviour only in the discussion section.

Response: We apologise from this confusion. This qualitative study was not informed by a theory or conceptual framework. We have clarified in the introduction that we used a qualitative study to explore adherence factors in this patient population, which could inform theory-led interventions on page 3.

“There is limited evidence on psychological and behavioural factors that support adherence to nutrition interventions for men with PC [20]. Furthermore, previous PA intervention studies with patients with chronic conditions, including cancer, have identified several factors that could support adherence to PA [21]. However, few of these studies have explored the psychological and behavioural factors which could align with exist models of behaviour change to enhance PA intervention in men undergoing prostatectomy. For example, a narrative review of behaviour change theories used in PA interventions in urological cancer survivors reported constructs of the Theory of Planned Behaviour (TPB) and the Trans-theoretical Model have been shown to increase men’s motivation to be more physical active either during or following PC treatment [22].

Our qualitative study aimed to identify factors associated with adherence to diet and PA interventions in men following prostatectomy for localised (organ-confined) PC, which could inform such theory-led interventions in this patient population.”

Second, the study design is not clearly defined. The authors mentioned that they performed a qualitative study as part of a secondary analysis of a factorial randomised controlled trial. Yet, there are several types of qualitative studies (See articles about this design written by M. Sandelowski in 2000 and 2010 in the journal *Research in Nursing & Health*). Therefore, the study type should be clearly specified.

Response: Thank you for the references on the types of qualitative studies. Our study was a “descriptive” qualitative study. This has been amended in the manuscript on page 3.

Third, it is unclear, what is the authors' argument for thematic analysis in the tradition of Braun and Clarke? Why is this method suited to the study at hand?

Response: We have provided a justification of our use of inductive thematic analysis in the data analysis section on page 5:

“Data were analysed using inductive thematic analysis with the aid of NVivo 10 software [26]. This method of analysis was chosen with the aim of understanding participant experiences of making behaviour changes beyond those related to study processes of the RCT (e.g., feasibility outcomes). There were also no preconceptions about what themes would be identified from the data.”

Forth, there are discrepancies between the study objectives and study methodology. The study objectives focus on the factors that supported and hindered men's adherence to the nutrition and physical activity; yet, in the method section on page 4 the authors specified that the participants “were asked mainly about their diet and PA before participation in the trial”.

We apologise for this confusion, which was due to grammatical error. We amended the respective paragraph on page 5 to clarify that we had one participant, who was in the control group, and only used data relating to his diet and PA before participation in the trial in the analysis:

“One man in the control group was included in the sampling. He received no intervention aside from standard publicly available nutrition and PA information, if requested. Data from response asked about his diet and PA before participation in the trial was only used for analysis.”

At the same time, the patient interview guide that the authors presented focuses mostly on the participants' experiences with the intervention, asking about the positives and negative elements of the intervention arm and participant's opinions about the associations between diet, physical activity, and cancer. There is unclear how the questions in the interview guide as shared in the paper helped answer the questions posed in the objective.

Response: Our objective was to identify factors that support behaviour change to the nutrition and physical activity interventions. As the interviews were semi-structured, men were able to elaborate on their responses to questions asked from the interview topic guide. The flexibility of this approach led to many of the men's responses extended beyond their experiences of performing the interventions from a trial perspective. As the analysis was data-driven, we were able to identify data that supported our study objective in the context of the men's daily life.

Fifth, at what level of abstraction did the authors aim in identifying themes? As I read all of them, each is purely descriptive and thus generally best classed as categories and not themes. Refer to the editorial, "Confusing Categories and Themes" by J. Morse (2008).

Response: Thank you for providing the reference to the editorial. Our analysis was descriptive in that we identified semantic (surface level) themes within the data. We analysed the data in line with Braun and Clarke (as referenced in the manuscript) who use the term ‘theme’. We, therefore, believe that we should continue using this term for consistency.

Might you revise your level of analysis? I also urge you to reconsider whether you have sufficient data to report certain themes.

Response: As mentioned above, our analysis was descriptive in that we identified surface level themes within the data. We acknowledge that the interviews were not in-depth as they were conducted as part of feasibility trial, which used the interviews to obtain data on trial

processes as well as intervention adherence. We are, therefore, unable to perform an in-depth analysis on the data. We also acknowledge that data was not as rich for some themes and we have mentioned this as a limitation on page 10:

“Data analysis was limited by the lack of depth in responses from the interviews. This is likely due to the interviews being part of data collection for a feasibility RCT, which assessed trial processes as well as intervention adherence.”

Most themes are described with a thin narrative and few quotes. The balance of narrative to data is typically indicative of the depth of development and representation of richness in the underlying data. Do the authors think they've adequately captured what the participants and data convey? I'd like to see a good deal more development here.

Response: As mentioned in our previous comment above, we acknowledge that the data was not in-depth and we have mentioned this in the discussion section on page 10. We chose the quotes which best supported each of themes. This led to some quotes not being as 'rich in qualitative data' as others. There were additional quotes that we could have included but did not do so for space reasons.

Sixth, it is important to include in the method and the discussion section the information on how the rigor of the study was established. There are a number of references for this, including the seminal text by Lincoln & Guba (1985) and the article, "Critical Analysis of Strategies for Determining Rigor in Qualitative Inquiry" by J. Morse (2015).

Response: Thank you for providing these reference by Lincoln & Guba (1985) and Morse (2015). We did perform negative case analysis and have amended the data analysis section to include this on page 5:

“Negative case analysis (i.e., identifying contradictory data) were used to broaden or confirm the interpretation of the themes and were resolved through discussion between the researchers and revisiting the transcripts.”

However, we were unable to perform other strategies to increase the rigour of the methods and have included the following text in limitations section on page 10:

“...this study has several limitations. Data analysis was limited by the lack of depth in responses from the interviews. This is likely due to the interviews being part of data collection for a feasibility RCT, which assessed trial processes as well as intervention adherence. Therefore, further assessments of rigour would not have benefitted the data analysis.”

We believe it is more appropriate to state this in the discussion rather than the methods section.

Seventh, there are multiple discrepancies between the statements (or the names of the categories) and quotes presented to support the statements/categories in the results section.

Responses: We have made amendments to the results section in accordance with your comments below. We believe this has resolved the discrepancies.

For instance, the author wrote on page 6: “While other men described making extensive changes on starting the PBD within the trial.” However, the quote that follows this statement is “As I say, I used to eat an awful lot of fruit and vegetables beforehand...”

Response: This quote referred to a participant who ate significant more portions of fruit and veg a day than advised. We considered this an extensive change to his diet even though he ate a lot of fruit and veg before the trial. We understand that the initial part of the quote confuses the point we were trying to make. Therefore, we have amended the quote as follows on page 6:

“...I found that I was really, sort of, [toning] myself up almost on fruit and veg. I think it said you had to eat 5 more portions of fruit and veg a day than normal, so I was getting up to, at some stages, about 20, I think, a day.” P17, PBD and brisk walking

Another example: the authors put the following statement as an example for the “physical limitations” category: “Most men also relied on good weather. There was also little motivation to walk elsewhere when the weather was bad. “...there were some days where it was a total wash-out, and you think, “Well, there’s no point in even trying,” you know. “I’ll make this my quiet day”...” However, it seems that the above-mentioned example is more appropriate for the “external obstacles category.”

Response: On review of the themes, we agree that this statement would be more appropriate in the external obstacles subtheme. We have, therefore, moved this statement to the external obstacles subtheme.

Furthermore, the results section is difficult to understand as it lacks a clear picture overall. Therefore, the result section will benefit if the thematic map of the study is presented.

Response: Thank you for the suggestion of a thematic map. We have now included one in the manuscript, which we believe offers a clearer picture (Figure 1).

Eighth, how are the authors constructing the discussion for this manuscript? Typically, a strong discussion recapitulates the main findings in brief and then goes on to argue original contributions, areas where findings corroborate extant literature, where the findings conflict, and finally limitations. This progression allows the authors to draw clearly defined conclusions in line with the method and results and to outline implications. I do not see this structure in your discussion.

Response: We agree that restructuring the discussion section would be beneficial. We have, therefore, rearranging some of the paragraphs in line with the order of themes presented in the results section. We have also included two additional subheadings (asterisked below) in the existing structure of the discussion listed below:

- *Summary of findings**
- *Support with other studies**
- *Strengths and limitations*
- *Main implications and future research*

Critically, many statements in the discussion section are not supported by the study result. For instance, the following statements in the discussion section are not supported clearly by the study results:

(1) “The findings suggest that a diagnosis of cancer can provide an opportunity for men to make changes to their nutrition and PA.”

Response: We agree that this statement does not fit well with the study results. We have, therefore, removed it from the discussion section.

(2) “While some men followed the intervention guidelines, others made quite extreme changes to their diet, such as eating well-over the recommended daily intake of fruit and vegetables.”

Response: We have commented about this point in the discussion section that men could benefit from more education on eating practices on page 9.

(3) “The facilitators to change identified in this study (i.e. ... self-efficacy with taking lycopene and exercising)”;

Response: We have amended the respective paragraph to support the study results on page 10:

“Our study findings indicate that men are motivated to make changes to their diet and level of PA following prostatectomy. However, men’s motivation was not related to beliefs that diet and PA was associated with their prostate cancer. Other psychological factors could explain men’s motivation to adherence to these behaviour changes, such as symptom control, which could be explored using

qualitative studies. Barriers to adhering to their behaviour changes related to physical (i.e., weather, time) and social opportunities (e.g., going on holiday). These findings suggest that future nutrition and PA interventions guided by a behavioural model, which help identify these barriers and incorporate techniques such as problem-solving, will improve adherence [41, 42]. The COM-B model [43] could be one that is suitable for this patient population. This model proposes that a person's motivation to perform and maintain a behaviour is supported by their capability (i.e., psychologically and physically) and opportunity (i.e., social and physical) to perform the behaviour. Future studies may consider exploring the use of this model in nutrition and PA intervention studies with PC populations."

(4) "This qualitative study suggests that behaviour change models could help both inform interventions and promote long-term adherence to nutritional and PA behaviours."

Response: We have corrected this sentence in the conclusion on page 11:

"This qualitative study suggests that behaviour change models could support adherence to nutritional and PA behaviours."

We also amended a sentence in the conclusion of the abstract on page 2:

"Future studies may benefit from theory-based interventions to support adherence to diet and PA behaviour changes in men diagnosed with prostate cancer."

Ninth, the "strengths and limitations" section should be revised. First, it is unclear how the study's strength could be its "findings on the psychological, behavioural, and social factors associated with adherence to men's changes to their diet and PA". Usually, the strength of the study should be judged on the rigor of the study methodology; therefore, the researchers should specify on what quality criteria they based their research. Please see, Lincoln & Guba (1985) and the article on "Critical Analysis of Strategies for Determining Rigor in Qualitative Inquiry" by J. Morse (2015), as well as, Treharne GJ, Riggs DW. Ensuring quality in qualitative research. In: Rohleder P, Lyons AC (eds) Qualitative Research in Clinical and Health Psychology. Basingstoke: Palgrave Macmillan, 2014; pp. 57–73.

Response: We have amended the strengths and limitations to relate specifically to the study methodology on page 10:

"The strength of this study is that it has provided a thematic analysis of men making diet and PA changes soon after prostatectomy, which included a negative case analysis to support the rigour of the study."

Furthermore, although in the result section the authors mentioned that "men across all the intervention arms believed that they maintained a healthy diet before being diagnosed" and the authors provided several quotes that described the men's healthy behaviors before the study, they did not mention this fact in the limitation section. However, it is well known that the new health-related behaviors' is easier to achieve in people with previous healthy behaviors compared to those without previous health behaviors; therefore, the factors that supported and hindered men's adherence to the nutrition and physical activity could be different among those with and without previous healthy behaviors.

Response: Thank you for highlighting this. We have added the following sentence to the limitations section on page 10:

"In addition, men in all the intervention arms discussed that they were already maintaining a healthy diet and engaging in regular physical activities before their diagnosis. This could suggest that the current findings are limited to men more willing and able to perform these health behaviours."

Reviewer: 2

Dr. Richard Crevenna, Medical University of Vienna

Comments to the Author:

To my opinion, the manuscript (bmjopen-2021-055566, "ATTITUDES AND ADHERENCE TO CHANGES IN NUTRITION AND PHYSICAL ACTIVITY FOLLOWING SURGERY FOR PROSTATE CANCER: A QUALITATIVE STUDY ") could be of interest of the reader of your journal, the BMJopen.

Importance of the question studied/described: adequate

Originality of work: adequate

Appropriateness of approach and experimental design: qualitative study using semi-structured interviews

Clarity of writing and soundness of organisation of the paper: adequate

Soundness of conclusion and interpretation and relevance of discussion: not adequate

Response: We believe the amendments we have made to the discussion has strengthened the conclusion and interpretation of the results.

Reference list: adequate

Reviewer: 3 [See attached file.]

Dr. Stefania Costi, University of Modena and Reggio Emilia

Comments:

General comments

The present manuscript is a qualitative study with the aim to explore factors that increase men's behavioural intentions and support their adherence to a diet and PA intervention. I think it could be of interest by readers of BMJ Open. However, there are some points, that I would ask you to clarify.

Special comments (as stated in the attached file)

Abstract

Page 4, line 12: I suggest writing the word physical activity in full for the first time and then using the abbreviation "PA".

Response: We have corrected this in the abstract on page 2.

Page 4, line 14: Could you explain what are the "theory-led interventions"?

Response: We have amended the respective sentence with the following on page 2:

"However, little is known about specific factors that support men's adherence to these health behaviour changes, which could inform theory-led diet and PA interventions."

Page 4, line 30-31: Are these barriers physical limitations (e.g. pain, constipation)? I suggest to specify what kind of nutrition and PA limitations and the external obstacles that men reported as a barriers.

Response: We believe this confusion is due the name of the theme (i.e., nutritional limitations). This theme describes men's ability to tolerate the changes to their diet. We have subsequently amended the theme to include both nutrition and PA interventions and renamed the theme "Tolerance to the intervention" on page 6.

Main text

Introduction

Page 5, line 7: I suggest to expand the rationale for PA and cancer survivorship. Exercise is stronger recommended as a strategy to prevent the side effects of PCa treatments and improve patients' quality of life. Two recent systematic reviews discussed the role of exercise in patients with prostate cancer receiving androgen deprivation therapy about its effectiveness, feasibility and safety (Bressi et al. Physical exercise for bone health in men with prostate cancer receiving androgen deprivation therapy: a systematic review. Support Care Cancer. 2021, doi: 10.1007/s00520-020-05830-1. Cagliari M et al, Feasibility and Safety of Physical Exercise to Preserve Bone Health in Men with Prostate Cancer Receiving Androgen Deprivation Therapy: A Systematic Review).

Response: Thank you for these references. We have added the references and amended our rationale for PA and cancer survivorship with the following on page 5:

“With regard to PA, observational studies suggest that moderate to vigorous PA is associated with reduced risk of PC-specific mortality and biochemical recurrence. More specifically, three hours of moderate to vigorous PA per week is associated with a 61% decrease in PC mortality compared with less than one hour [9]. The increase of PA on lower risk of PC-specific mortality and recurrence is supported by intervention studies [10]. In addition, PA has been shown to reduce adverse effects of treatment and improve quality of life, particular in men receiving androgen derivation therapy [11, 12].”

Methods

Page 5, line 41: Was the study prospectively recorded in one of the available public databases?

Response: This qualitative study was part of the Prostate cancer – Exercise and Nutrition Trial, which was retrospectively registered on the ISRCTN registry. We have added the registry number (ISRCTN99048944) to the methods section on page 3.

Page 6, line 12: How were participants approached (e.g. face-to-face, telephone, email...)?

Response: Participants were approached in person. We have stated this on page 4.

Page 6, line 12: Could you specify the “external circumstances” of the six men unable to attend?

Response: These have been added to the text on page 4:

“Six men were unable to attend due to external and personal circumstances (i.e., did not have the time during the clinic appointment (n=3), interviewer not available (n=2), and participant unwell (n=1))”

Page 6, Table 2: Could you specify in the table legend what means “further education”, with examples?

Response: We have added examples next to ‘further education’ in the table. We have also added examples to ‘secondary school’ for consistency.

Page 6, line 45: What was the duration of the interviews?

Response: We have added that the interviews lasted between 19 and 84 minutes on page 5.

Page 6, line 46: Why did you decide to not include focus group to encourages respondents and interaction with each other, and to share perspectives?

Response: We did not use focus groups as interviews were conducted immediately or shortly after participants had finished their interventions to reduce recall bias. Participants were required to consent to be interviewed. It was much easier to do this whilst they were present in the clinic and reduced missing participants.

Page 6, line 49: What were the researchers credentials? What experience or training did the researchers have?

Response: We have added the researchers' background to the methods section on page 5:

"Interviews were conducted by three authors (ES, n=9; LM, n=7; LR, n=1), whose backgrounds include Public Health (ES) and Health Psychology (LM, LR)..."

Page 6, line 51: By whom and how was the questionnaire written (Supplementary material 1)? Were questions, prompts, guides provided by the authors?

Response: The interview topic guide (Supplementary material 1) was created by the authors and was reviewed by a PPI group before the interviews were conducted. This is mentioned page 5 in the manuscript.

Page 6, line 51: What methodological orientation was used (e.g. grounded theory, discourse analysis, ethnography, phenomenology, content analysis)?

Response: We used inductive thematic analysis and identified semantic themes across the transcripts. We have clarified this in the manuscript on page 5.

Page 6, line 53-54: I understand that these sentences referred to the man (n = 1) included in the control group of the RCT Prostate cancer Evidence of Exercise and Nutrition Trial (PrEvENT), but the subject is plural. Please clarify.

Response: Thank you for spotting this error. We have corrected these sentences with the following on page 5:

"One man in the control group was included in the sampling. He received no intervention aside from standard publicly available nutrition and PA information, if requested. Data from responses about his diet and PA before participation in the trial was only used for analysis."

Results

Page 7, line 39: I suggest to specify only the first time that PBD is the Plant-based diet, and then use the acronym consistently.

Response: We state that PBD is the Plant-based diet earlier in the manuscript on page 3 and then use the acronym throughout the manuscript, except in the tables. We have, therefore, made no changes regarding this comment.

Page 7, line 23: It could be of interest a description of some theme of the man included in the control group of the PrEvENT study.

Response: Data from the man in the control group was included in the analysis of the first two themes (i.e., Causal beliefs about prostate cancer, Perceptions of a healthy diet and PA before diagnosis). We do not believe adding quotes from this man would add anything further to these themes.

Discussion

Page 11, line 32: In the literature the role of partner is considered a usefull support for men with PCa to adopt and maintain healthy behaviours. Partner may influence each other's diet and also exercise behaviours (PMID: 25807856). Your findings did not confirm the role of partners in supporting PA, so I suggest to discuss this point.

Response: We have now discussed our finding of partner involvement in PA interventions in the discussion on page 9:

“Partners were found to be significantly involved in choosing and preparing meals for men. Partners are often involved at each stage of men’s treatment pathway, including helping them comply with pre-prostatectomy preparation, such as improving fitness and losing weight [34]. Thus, this finding suggests that men would adhere better to PBD interventions with partner involvement. In contrast, men discussed the PA intervention as one which they preferred to do by themselves. This finding is supported by existing studies in which some men with prostate cancer have discussed partners being a facilitator in PA intervention while others preferred their partners not to be involved and to exercise with men only [35].”

Reviewer: 4

Ms. Kim Edmunds, Griffith University

Comments to the Author:

Thankyou for the opportunity to review this paper. Lifestyle compliance and adherence are important considerations for cancer patients and it is pleasing to see qualitative investigation happening alongside randomised clinical trials. I thoroughly enjoyed the paper and encourage you to extend your research in this important area as you propose in your protocol.

While much research has been carried out on adherence, PCa patients with early stage disease are a unique group because they tend to be relatively healthy and the evidence for the impact of lifestyle change on their disease is strong. Any research that interrogates the why behind adherence or otherwise is always welcome. That said, there are areas in your paper I believe could be strengthened. Most importantly is that much of the research that backgrounds your study tends to be quite dated. It gives the reader the impression that you are not up to date with the latest developments in exercise oncology. International leaders in this area in PCa are Robert Newton Daniel Galvao, Dennis Taaffe, Kerry Courneya. Other exercise oncologists of international note are Kathryn Schmitz, Anna Campbell, Sandi Hayes, and others.

Response: Thank you for your helpful comment. We have provided a up to date references in the background section on page 3 (reference numbers: 8, 10, 11, 12, 20, 21, 22).

Thank you for incorporating a PPI group in your research.
I comment on the paper as and where such issues arise.

Abstract Page 4

Line 22-expression and accuracy

Participants were 17 men with a median age of 66 years,,who underwent surgery
Why don't you use prostatectomy rather than surgery, as you did in you feasibility study?
I would think it more appropriate for your readership at BMJ.

Response: We have changed surgery to prostatectomy.

Line 28-expression

relationship of nutrition and PA with PCa...OR the impact of nutrition and PA on PCa...

Response: We have changed relationship to impact.

Page 5

Line 5-expression

There is also evidence that high intake of...

Response: We have amended this sentence on page 3:

“High intakes of dairy products is also associated with increased prostate cancer risk [8, 9].”

Line 9ff updated references

Many of the references cited are somewhat dated and while Kenfield is a seminal article, systematic reviews by internationally recognized exercise physiologists (e.g. Cormie et al. 2017 The impact of exercise on cancer mortality...)have updated these and are worth including.

Response: Thank you for the reference. We have added it in the introduction section on page 3.

Line 19 expression

...were meeting the recommendations for fruit...

OR rewrite

...out of 2000 PCa survivors, only 43% were meeting recommendations for fruit and vegetable consumption and only 16% (were meeting recommendations) for PA.

Response: We amended this sentence with the following on page 3:

"... only 43% were meeting the recommendations for fruit and vegetable consumption and only 16% were meeting the recommendations for PA."

Line 27

This reference is 2011. Much has been written since about this area re motivation, compliance and adherence in exercise oncology. Courneya, K. did much of the early work. A recent umbrella review which includes seven exercise oncology papers is informative in terms of all identifying the key factors (Collado-Mateo 2021 Key factors associated with adherence to physical exercise..). I believe you need to situate your study within this more recent literature.

Response: Thank you for providing this reference. We have added it to our introduction on page 3.

Page 6

Line 15 redundant language

...reported as White British...

Response: This has been corrected.

Line 49 consistent use of numbering

...All three authors...

Response: This has been corrected.

Page 10

Line 6 expression

...others, most men claimed/admitted that..,

Response: We have changed the sentence to: "...others, most men claimed that..."

More recent research needs to be incorporated in your discussion. This is where you compare what you found with what others have found, so ideally, should be the most recent research in the area.

Response: Thank for your helpful comment. We have provide more recent citations to support our findings with other studies where appropriate (reference numbers: 28, 33, 36, 39)

Page 11

Line 28 verb form consistency

...three constructs: having a positive attitude to a behaviour, (2) perceiving others to be supportive of it, and (3) believing...

Response: We have corrected this sentence.

Line 57

...another study

Response: This has been corrected.

Page 12

Line 3 ...another qualitative study...

Response: This has been corrected.

Interestingly, both the findings from these studies seem to suggest that these men prefer to blame things outside their control or not take responsibility for their actions (environment and work/life stress, vs overeating and being inactive?? Perhaps this could be used in devising motivational change...??? Just a thought

Response: Thank you for this helpful comment. We had added this point to our discussion section on page 9:

“Men were not fully convinced that cancer was caused or related to their nutrition or PA. They attributed the cause of their cancer to external factors including age and genetic factors. These findings are supported by another qualitative study [27], which shown that prostate cancer survivors can overestimate the significance of environmental factors, such as pollution and stress, and underestimate behaviour factors associated with increased cancer risk, such as obesity and inactivity. In contrast, a systematic review examining cancer beliefs in relation to fear of recurrence found positive associations between both internal and external causal beliefs (i.e., diet, hormones) and engagement with diet and PA among women with breast cancer and gynecological cancer [28]. These findings could be indicative of men’s preference to believe in causal factors that are outside their control.”

I am not sure I agree with your tentative finding that men would adhere to healthy changes regardless of their perceived risk. Perhaps something else is happening that fits more with their preference not to take responsibility...if some one organises it and tells them what to do, they will happily do it??? Just a thought

Response: Thank you for this helpful comment. We have added this point to the Implications and future research section on page 10:

“Our study findings indicate that men are motivated to make changes to their diet and level of PA following prostatectomy. However, men’s motivation was not related to beliefs that diet and PA was associated with their prostate cancer. Other psychological factors could explain men’s motivation to adherence to these behaviour changes, such as symptom control, which could be explored using qualitative studies.”

I think the small sample size is a limitation that should be mentioned.

Response: Thank you for this suggestion. We have added the small sample size as a limitation on page 10.

Congratulations on a well conducted study on an important topic. My suggested changes are minor. However, I believe it is important in the background and in the discussion to incorporate other more recent studies against which to compare your own.

Response: Thank you for your helpful comments. We believe we have strengthened the background and discussion sections using your comments.

All the best with your future research.

REVIEWER	Costi, Stefania University of Modena and Reggio Emilia
REVIEW RETURNED	16-Mar-2022

GENERAL COMMENTS	Dear Author of the manuscript "ATTITUDES AND ADHERENCE TO CHANGES IN NUTRITION AND PHYSICAL ACTIVITY FOLLOWING SURGERY FOR PROSTATE CANCER: A QUALITATIVE STUDY", I have read with interest your manuscript which, in my opinion, gives interesting information regarding the perspectives of men with prostate cancer regarding the benefit of adopting a healthy lifestyle. Although I have limited experience in qualitative research, I understand that the research has been conducted with rigor. I did not review the work the first time but it seems to me that you have made numerous changes that make the manuscript clearer and more complete for the readers. The only thing I ask of you is a very minor revision, that is to integrate into the discussion the results of a very recent study that has several points in common with your results, and can therefore support your arguments under discussion. This is the study by Bressi and colleagues published in February 2022 (PMID: 35194723), the results of which say that a high proportion of men are insufficiently active when diagnosed with PCa. More than 60% are obese. However, even when exposed to behavioral risk factors, they are not willing to change their lifestyle. This seems to confirm the incomplete awareness of the role that lifestyle has in the genesis of cancer and on health in general. I would suggest to add this very recent reference in the paragraph "Support with other studies", because it can support the statement that you write in that paragraph. The same study also reports barriers and facilitators to exercise, confirming that lack of time and bad weather act like barriers while the perception of a psychological and physical benefit associated to exercise are facilitators. Thus, I believe these results (obtained in a cohort of 40 men with PC) can make your results stronger.
--

VERSION 2 – AUTHOR RESPONSE

Reviewer: 3

Dr. Stefania Costi, University of Modena and Reggio Emilia

Comments to the Author:

Reviewer comment: Dear Author of the manuscript "ATTITUDES AND ADHERENCE TO CHANGES IN NUTRITION AND PHYSICAL ACTIVITY FOLLOWING SURGERY FOR PROSTATE CANCER: A QUALITATIVE STUDY", I have read with interest your manuscript which, in my opinion, gives interesting information regarding the perspectives of men with prostate cancer regarding the benefit of adopting a healthy lifestyle. Although I have limited experience in qualitative research, I understand that the research has been conducted with rigor.

Authors' response: Thank you for your positive comments.

Reviewer comment: I did not review the work the first time but it seems to me that you have made numerous changes that make the manuscript clearer and more complete for the readers. The only thing I ask of you is a very minor revision, that is to integrate into the discussion the results of a very

recent study that has several points in common with your results, and can therefore support your arguments under discussion. This is the study by Bressi and colleagues published in February 2022 (PMID: 35194723), the results of which say that a high proportion of men are insufficiently active when diagnosed with PCa. More than 60% are obese. However, even when exposed to behavioural risk factors, they are not willing to change their lifestyle. This seems to confirm the incomplete awareness of the role that lifestyle has in the genesis of cancer and on health in general. I would suggest to add this very recent reference in the paragraph "Support with other studies", because it can support the statement that you write in that paragraph.

Authors' response: Thank you for suggesting this very interesting reference. We have added the study by Bressi and colleagues (reference 29) in the paragraph below the subheading 'Support with other studies' on page 9 along with the following text:

"Furthermore, in a sample of 40 men interviewed about their lifestyle behaviours following their PCa diagnosis, 60% were obese and 88% were not motivated to change their smoking, alcohol and/or their eating behaviours [29]. These findings could be indicative of men's preference to believe in causal factors that are outside their control, and reinforce the importance of lifestyle interventions at the time of diagnosis."

Reviewer comment: The same study also reports barriers and facilitators to exercise, confirming that lack of time and bad weather act like barriers.

Authors' response: We have added the reference (29) on page 9 to support these findings.

Reviewer comment: While the perception of a psychological and physical benefit associated to exercise are facilitators. Thus, I believe these results (obtained in a cohort of 40 men with PC) can make your results stronger.

Authors' response: We have added the reference (29) on page 10 and amended the sentence (before the strengths and limitations section) with the following:

"Such physical and psychological outcomes have been reported as facilitators to exercise [29] and have the potential to help men adhere well to their brisk walking."

Reviewer: 3

Competing interests of Reviewer: None

Authors' minor amendment: We have changed the acronym for prostate cancer (PC) to PCa as this acronym is more widely recognised. We also made a minor change to the last sentence in the Acknowledgements section with the following:

"The authors would also like to thank Dr Aidan Searle for his advice on the qualitative analysis of this study."